# Se-Doped Ni₅P₄ Nanocatalysts for High-Efficiency Hydrogen Evolution Reaction

Cuihua An [1,2], Yuchen Wang [2], Penggang Jiao [1], Shuai Wu [1,*], Lingxiao Gao [1], Chunyou Zhu [3], Junsheng Li [4,*] and Ning Hu [5]

[1] Key Laboratory of Hebei Province on Scale-Span Intelligent Equipment Technology, School of Mechanical Engineering, Hebei University of Technology, Tianjin 300401, China

[2] Institute for New Energy Materials & Low-Carbon Technologies, School of Materials Science and Engineering, Tianjin University of Technology, Tianjin 300384, China

[3] Hunan Aerospace Kaitian Water Services Co., Ltd., Changsha 410100, China

[4] School of Chemistry, Chemical Engineering and Life Sciences, Wuhan University of Technology, Wuhan 430070, China

[5] State Key Laboratory of Reliability and Intelligence Electrical Equipment, Hebei University of Technology, Tianjin 300130, China

* Correspondence: wushuai@hebut.edu.cn (S.W.); li_j@whut.edu.cn (J.L.)

**Abstract:** Increasing energy consumption and environmental pollution problems have forced people to turn their attention to the development and utilization of hydrogen energy, which requires that hydrogen energy can be efficiently prepared. However, the sluggish kinetics of hydrogen evolution reaction (HER) requires higher overpotential. It is urgent to design and fabricate catalysts to drive the procedure and decrease the overpotential of HER. It is well known that platinum catalysts are the best for HER, but their high cost limits their wide application. Transition metals such as Fe, Co, Mo and Ni are abundant, and transition metal phosphides are considered as promising HER catalysts. Nevertheless, catalysts in powder form are very easily soluble in the electrolyte, which leads to inferior cycling stability. In this work, Ni₅P₄ anchored on Ni foam was doped with Se powder. After SEM characterization, the Ni₅P₄-Se was anchored on Ni foam, which circumvents the use of the conductive additives and binder. The Ni₅P₄-Se formed a porous nanosheet structure with enhanced electron transfer capability. The prepared Ni₅P₄-Se exhibited high electrochemical performances. At 10 mA cm$^{-2}$, the overpotential was only 128 mV and the Tafel slope is 163.14 mV dec$^{-1}$. Additionally, the overpotential was stabilized at 128 mV for 30 h, suggesting its excellent cycling stability. The results show that Se doping can make the two phases achieve a good synergistic effect, which makes the Ni₅P₄-Se catalyst display excellent HER catalytic activity and stability.

**Keywords:** HER; Ni₅P₄; Se-doped; electrocatalyst; electron transfer

## 1. Introduction

As an energy carrier with high specific energy density and zero greenhouse gas emissions, hydrogen can be used as a clean renewable energy on a large scale [1–5]. However, there is a high overpotential in the process of electrolysis of water, resulting in a large amount of energy consumption. It is necessary to use efficient catalysts to reduce this overpotential [6–9]. At present, the catalyst with the best hydrogen evolution performance is platinum group metals, but the scarcity and high price make large-scale application difficult [10–16].

Nickel is widely known for its excellent HER performance. Therefore, the enthusiasm of researchers to improve the performance of nickel-based catalysts in alkaline solution has not subsided [17,18]. The surface adsorption properties of nickel-based alloys and heterostructure catalysts can optimize their electrocatalytic activity and stability by fine-tuning the synergistic effect generated by adjacent elements [19,20]. Extensive work has found that

phosphides have stable behavior in acidic and basic solutions and high current densities at low overpotentials, which make them show great potential as HER catalysts [21–23]. The phosphorus in the metal phosphide structure is moderately bonded to the reaction intermediate to participate in the reaction, resulting in the formation of surfaces with proton acceptor and hydride acceptor sites. These reasons make metal phosphides highly active [24–26]. Ni and P elements are abundant in the crust. Nickel phosphide compounds have exhibited good electrochemical hydrogen evolution performance in the previous reports [27]. Among them, $Ni_5P_4$ particles reveal superior HER performance in both acidic and alkaline solutions [27–29]. The preparation of these catalysts is mostly based on nanoparticles. Despite the large surface area of nanoparticles, issues such as uncontrolled agglomeration, high series resistance and susceptibility to oxidation are also detrimental to the overall performance. $NiSe_2/Ni_5P_4$ nanosheets grown in situ on nitrogen-doped carbon nanofibers were prepared for water splitting [30]. $NiSe_2/Ni_5P_4$, which has a unique porous nanostructure, was obtained by low-temperature phosphating/selenization of $Ni(OH)_2$. In $H_2SO_4$ solution, the HER performance of $NiSe_2/Ni_5P_4$ is excellent, with an overpotential of only 112 mV at 10 mA cm$^{-2}$. The heterostructure of $NiSe_2/Ni_5P_4$ optimizes the adsorption of hydrogen-containing intermediates, which enables the excellent HER activity of $NiSe_2/Ni_5P_4$ on nitrogen-doped carbon nanofibers. Zhuo et al. obtained a HER catalyst grown on carbon paper by doping pyrite-phase nickel phosphide with Se [31]. Compared with undoped $NiP_2$, the overpotential and Tafel slope of Se-doped $NiP_2$ ($NiP_{1.93}Se_{0.07}$) have obvious advantages. The above results show that Se doping can improve the HER performance of $NiP_2$. The two phases of biphasic catalysts are promisingly synergistic to optimize the adsorption–desorption behavior of intermediates at active sites [30,32,33]. Nevertheless, the catalyst in powder form is very easily soluble in the electrolyte, which leads to the inferior cycling stability. Moreover, the addition of the conductive additives and binder increases the whole cost.

Here, we used Ni foam as the substrate and sodium hypophosphite as the phosphorus source to prepare a $Ni_5P_4$ anchored on Ni foam for water electrolysis through a gas-phase reaction with a low-process cost. The structure, morphology and hydrogen evolution properties were characterized. After that, we tried to use selenium powder as the selenium source to dope the catalyst to form the $Ni_5P_4$-Se composite, which further reduced the overpotential. Among the four catalysts, the $Ni_5P_4$-Se anchored on Ni foam displayed the best HER performances. At 10 mA cm$^{-2}$, the $Ni_5P_4$-Se composite displayed an overpotential of only 128 mV and possessed a Tafel slope of 163.14 mV dec$^{-1}$. The $Ni_5P_4$-Se electrode also exhibited excellent cycling durability, with an overpotential stabilized at 128 mV when tested at 10 mA cm$^{-2}$ for 10 h.

## 2. Experimental

### 2.1. Synthesis

The synthesis of the $Ni(OH)_2$ precursor: First, the Ni foam was divided into small pieces (1 × 1.5 cm), soaked in 1 M HCl solution for 15 min, and ultrasonically cleaned with ultrapure water and anhydrous ethanol for three times. Then, 6 mmol $Ni(NO_3)_2 \cdot 6H_2O$ and 10 mmol urea were dissolved in 30 mL of deionized water, and a clear and transparent green solution was obtained after magnetic stirring for 30 min. The treated Ni foam was immersed in the green solution. The above solution was transferred to an autoclave, which was placed in an oven at 180 °C for 12 h. After cooling down to the room temperature, the sample was taken out and washed with ultrapure water for several times. Lastly, the $Ni(OH)_2$ precursors in situ grown on the Ni foam were obtained.

The synthesis of the $Ni_5P_4$-Se nanocatalysts: The $Ni(OH)_2$ precursor was placed on the downstream side of the tube furnace. In all, 1 g $NaH_2PO_2 \cdot H_2O$ and 50 mg Se powder were placed on the upstream side. Then, the tube furnace was heated to 300 °C at a heating rate of 2 °C min$^{-1}$ under an argon atmosphere, which was kept for 120 min. After naturally cooling to room temperature, in situ $Ni_5P_4$-Se nanocatalysts grown on nickel foam were obtained. Single-phase $Ni_5P_4$ and $NiSe_2$ were prepared by the above-mentioned method

from equimolar amounts of $NaH_2PO_2 \cdot H_2O$ and Se powder as phosphorus and selenium sources, respectively.

## 2.2. Characterization

The physicochemical properties of the catalysts were mainly characterized using the following means. The microstructure, morphology and elemental constitution of samples were characterized on field emission scanning electron microscope (SEM, Quanta FEG 250), super-resolution SEM (Verios 460L, Hillsboro, OR, USA), transmission electron microscope (TEM, $LaB_6$) and high-resolution TEM (Talos F200X, Hillsboro, OR, USA). The X-ray diffraction (XRD) patterns of the materials were obtained on a MiniFlex600 X-ray diffractometer (Rigaku, Japan). The scan rate is $5° \ min^{-1}$. The range is from $10°$ to $90°$. An X-ray photoelectron spectrometer (XPS, ESCALAB 250Xi, Waltham, MA, USA) was used to determine the elemental composition and chemical valence of the samples. The samples for SEM, XRD and XPS characterizations are powder. For TEM characterization, the powder sample is dispersed in ethanol to form a homogeneous solution, which is dropwise added into Cu mesh.

## 2.3. Electrochemical Measurements

The electrochemical measurements were conducted on the CHI660E electrochemical workstation. In the three-electrode system, the prepared material was used as the working electrode, graphite as the counter electrode and saturated calomel electrode (SCE) as the reference electrode. The measured raw data potentials were relative to the SCE. They needed to be converted into a standard reversible hydrogen electrode (RHE) potential. The specific conversion formula is: E (vs. RHE) =E (vs. SCE) + (0.059 × pH) +0.241 V. Electrochemical measurements include: linear sweep voltammetry (LSV), cyclic voltammetry (CV), electrochemical impedance spectroscopy (EIS) and galvanostatic stability curve (V-t).

## 3. Results and Discussion

$Ni(OH)_2$ anchored on Ni foam was prepared by hydrothermal method. As illustrated in Figure S1, the diffraction peaks of the $Ni(OH)_2$ catalyst are very obvious and match well with JCPDS No. 74-2075. It is proved that the pure phase $Ni(OH)_2$ was successfully obtained with good crystallinity. The $Ni(OH)_2$ nanoarrays are relatively uniformly grown on the ligaments surface of the Ni foam (Figure S2a). The grown $Ni(OH)_2$ precursor is a flower-like cluster structure composed of many nanosheets, which can be clearly observed in the local enlarged view of the Ni foam ligament (Figure S2b). Moreover, the content of $Ni(OH)_2$ in the composites is about 24 wt%. The $Ni(OH)_2$ precursor was subjected to phosphorization, selenization or simultaneous phosphorization and selenization reactions. The XRD patterns of the obtained products are shown in Figure 1a. The diffraction peaks of single-phase $Ni_5P_4$ and $NiSe_2$ are matched with JCPDS No. 28-0883 and JCPDS No. 41-1495, respectively, indicating that the $Ni(OH)_2$ precursor was completely converted into $Ni_5P_4$ and $NiSe_2$. The XRD pattern of the $Ni_5P_4$-Se is almost close to that of the single-phase $Ni_5P_4$ (Figure 1a). No obvious selenium diffraction peaks were presented in the XRD pattern of the $Ni_5P_4$-Se due to the relatively low content of the Se element source. The doping of Se induces the formation of a two-phase heterojunction and reduces the crystallinity of the single-phase $Ni_5P_4$. Therefore, by observing the characteristic peaks of the $Ni_5P_4$ phase, it can be found that Se doping significantly weakens the peak intensity of the $Ni_5P_4$ phase. The corresponding microstructures were also characterized. It can be clearly observed in Figure S3a,b that a single $Ni_5P_4$ nanoarray grows uniformly on the Ni foam. Many active sites cannot be exposed in Figure S3c because of the accumulation of a large number of large sheets to form clusters or particles. The morphology of the layered $NiSe_2$ array is displayed in Figure S4. The $NiSe_2$ nanosheets grow more densely, which possess an approximate microstructure with the $Ni(OH)_2$ precursor.

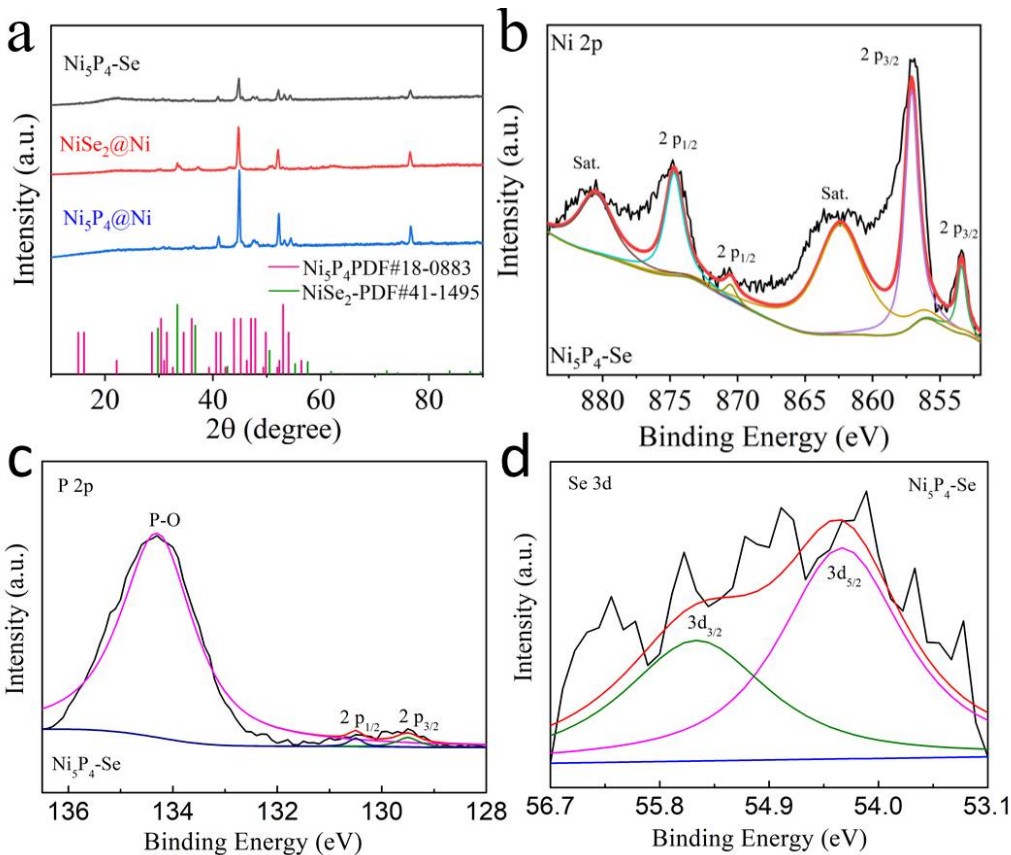

**Figure 1.** (**a**) XRD patterns of $Ni_5P_4$, $NiSe_2$ and $Ni_5P_4$-Se. XPS spectra of $Ni_5P_4$-Se: (**b**) Ni 2p, (**c**) P 2p, (**d**) Se 3d.

The high-resolution XPS spectras of $Ni_5P_4$ and $NiSe_2$ are included in Figure S5. The Ni 2p spectrum of $Ni_5P_4$ in Figure S5a illustrates that the peaks located at 853.4 eV and 870.5 eV correspond to Ni $2p_{3/2}$ and Ni $2p_{1/2}$, respectively. The peaks centered at 857.1 eV ($2p_{3/2}$) and 874.7 Ev ($2p_{1/2}$) can be ascribed to the oxidation state of Ni. Two doublets are located at 129.6 eV and 130.4 eV in the P 2p spectrum, which belong to P $2p_{3/2}$ and P $2p_{1/2}$, respectively. The characteristic peak at 134.2 eV suggests the existence of the P–O bond. There are two peaks at 855.8 eV and 873.1 eV in the Ni 2p spectrum of $NiSe_2$, which are ascribed to Ni $2p_{3/2}$ and Ni $2p_{1/2}$, respectively (Figure S5c). The peak at 854 eV ($2p_{3/2}$) corresponds to nickel oxide. In addition, two satellite peaks were observed at 855.8 eV and 879.5 eV. There are two peaks here at 55.1 eV and 54.5 eV in the Se 3d spectrum corresponding to Se $3d_{3/2}$ and Se $3d_{5/2}$, respectively. (Figure S5d).

The XPS measurement of the $Ni_5P_4$-Se was conducted to investigate the chemical composition and valences (Figure 1b–d). Similarly, several deconvoluted peaks (853.4 eV for Ni $2p_{3/2}$ and 870.5 eV for Ni $2p_{1/2}$) were obtained in C 1s spectra (Figure 1b). The peaks at 857.1 eV ($2p_{3/2}$) and 874.7 eV ($2p_{1/2}$) can be owing to the oxidation state of Ni. The two doublets at 129.6 eV and 130.4 eV in P 2p spectra are attributed to P $2p_{3/2}$ and P $2p_{1/2}$, respectively (Figure 1c). The characteristic peak at 134.2 eV is ascribed to the P–O bond. In the Se 3d spectrum shown in Figure 1d, the peaks located at 54.9 eV and 54.2 eV belong to Se $3d_{3/2}$ and Se $3d_{5/2}$, respectively. Compared with the single-phase $Ni_5P_4$, the P 2p peak exhibits a small positive shift (0.4 eV) after Se doping. In contrast, the two peaks of Se $3d_{5/2}$ and $3d_{3/2}$ are negatively shifted (0.2 eV) compared to the single-phase $NiSe_2$. This phenomenon demonstrates that electrons are transferred from P to Se at the locally formed heterostructure interface, suggesting the formation of a strong internal electronic effect heterointerface. These will lead to the transfer and redistribution of electrons. Because of

this electronic effect, the heterointerface provides more reactive sites, reduces the catalytic reaction overpotential and improves the catalytic efficiency.

The nanosheet layer uniformly grown on the ligament surface of the Ni foam was obtained through the steam reaction jointly driven by sodium hypophosphite and selenium powder (Figure 2a,b). The partial enlarged view of the nanosheet layer is displayed in Figure 2c. A regular nanosheet array with a size of about 500–1000 nm can be observed in the local area. Since these regular nanosheet arrays are not similar to other large-scale curved nanosheets, we infer that the $Ni_5P_4$/$NiSe_2$ heterojunction structure is locally grown on the ligaments of the Ni foam. Energy Dispersive Spectrometer (EDS) and elemental analysis were performed on this region to further analyze the doping situation of Se. In the regular nanosheet region in Figure 2d–f, we can clearly observe that the content of the Se element is higher than that of the P element, while the P element is dominant in the large-sized nanosheet region. The morphological structure of the $Ni_5P_4$-Se composites were verified by the TEM images (Figure S6). From the above results, it can be roughly concluded that the Se-doped $Ni_5P_4$ phase is mainly concentrated in this region. The synergistic effect of the regular two-phase structure nanosheets provides more active sites for HER. The Energy Dispersive Spectrometer (EDS) spectrum is shown in Figure S7. The ratio of the Se and P atoms in this region is about 1:1, which further verifies the successful doping of Se into $Ni_5P_4$.

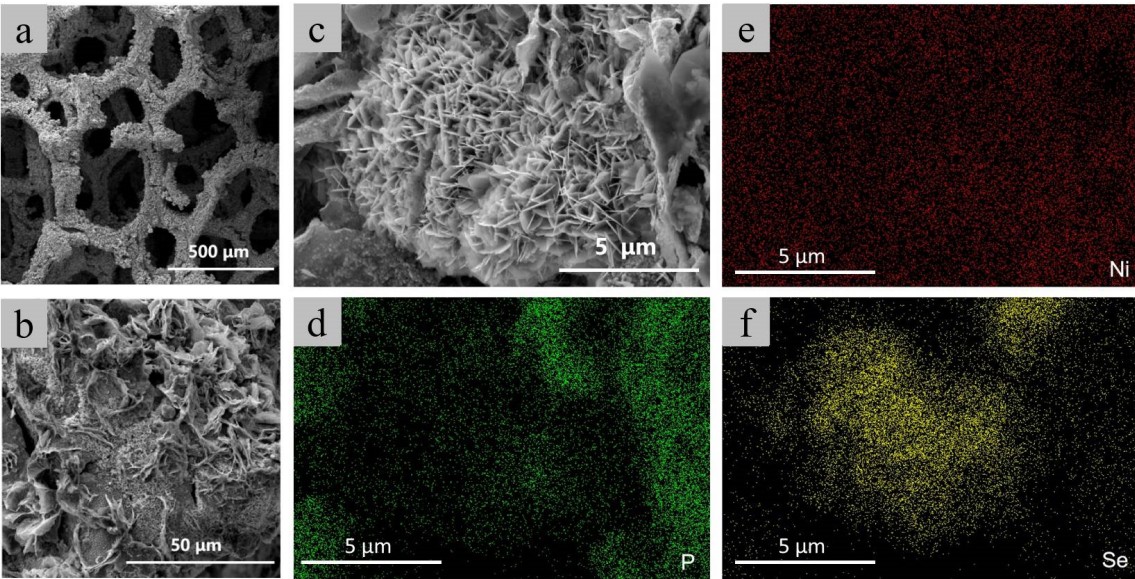

**Figure 2.** Low magnification SEM images (**a–c**) and corresponding elemental distributions of $Ni_5P_4$-Se composites (**d–f**).

The electrochemical HER performances of the $Ni(OH)_2$ precursor, the single-phase $Ni_5P_4$ and $NiSe_2$ and the $Ni_5P_4$-Se were measured using a three-electrode system. The LSV curves of the four catalysts at 10 mVs$^{-1}$ are depicted in Figure 3a. The overpotential of the the $Ni_5P_4$-Se is only 128 mV at 10 mA cm$^{-2}$, which is significantly better than that of the single-phase $Ni_5P_4$ (150 mV), $NiSe_2$ (211 mV) and $Ni(OH)_2$ (278 mV). To explore the reaction kinetics in depth, we fitted the Tafel slope according to the Tafel equation. The Tafel slope of the $Ni_5P_4$-Se electrocatalyst is 163.14 mV dec$^{-1}$ in Figure 3b, which has obvious advantages compared with $Ni_5P_4$ (177.94 mV dec$^{-1}$), $NiSe_2$ (183.94 mV dec$^{-1}$) and $Ni(OH)_2$ (208.77 mV dec$^{-1}$). The $Ni_5P_4$-Se catalysts exhibited good reaction kinetics and a higher electron transfer efficiency, which makes the catalytic reaction easier. Meanwhile, in order to explore the charge transfer kinetics, the electrochemical impedance spectra (Figure 3c) show the charge transfer resistance of the four catalysts. The semicircle of the Nyquist plot refers to the charge transfer resistance Rct during the HER process. Obviously, the electrocatalyst of the $Ni_5P_4$-Se exhibits the smallest semicircular diameter, which further

proves that the doping of Se accelerates the electron transfer of the single-phase $Ni_5P_4$ and the HER catalytic kinetics are stronger. The long-cycle stability of the $Ni_5P_4$-Se electrode was measured at 10 mA cm$^{-2}$ for 30 h (Figure 3d). The overpotential was stabilized at 128 mV, demonstrating its excellent cycling durability. Moreover, the SEM image of the $Ni_5P_4$-Se catalyst is displayed in Figure S8. As we can see, the $Ni_5P_4$-Se catalyst exhibits a nanosheet structure, which maintains its original morphological structure, further demonstrating its structural stability. The comparison of the catalytic activity of catalysts is presented in Table S1 [34–43].

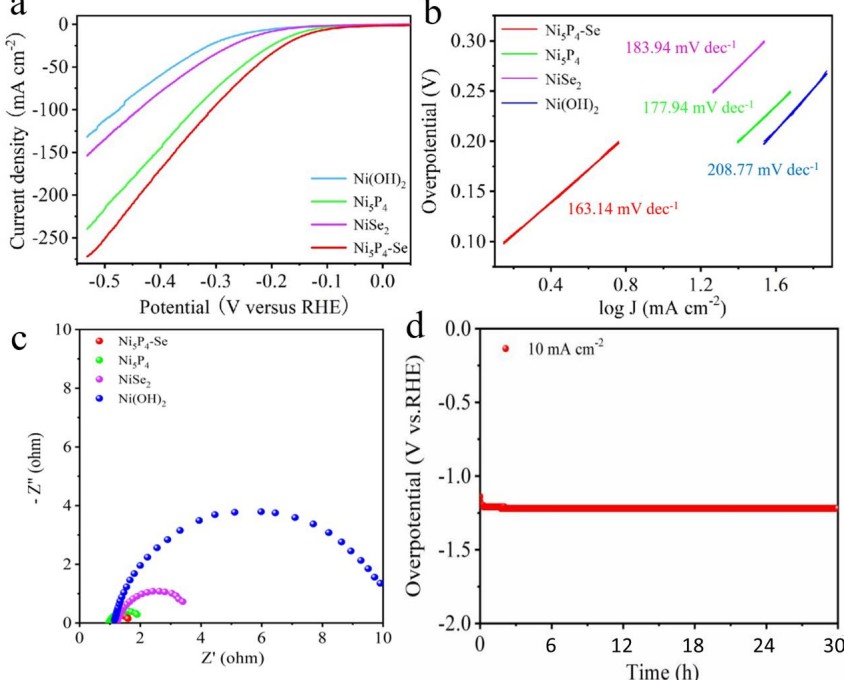

**Figure 3.** (**a**) Polarization curve, (**b**) Tafel slope, (**c**) Nyquist plots of $Ni(OH)_2$, $Ni_5P_4$, $NiSe_2$, $Ni_5P_4-$Se samples measured in 1 M KOH electrolyte., (**d**) V-t curves of $Ni_5P_4-$Se at 10 mA cm$^{-2}$.

An electrochemically active surface area (ECSA), which was evaluated by measuring electrochemical double layer capacitance ($C_{dl}$), was used to further study the electrocatalytic kinetic performance of the catalyst. The Faradaic potential region was selected as shown in Figure 4. The CV curves were tested at 10, 20, 50, 80 and 100 mV s$^{-1}$, respectively. The ECSAs of the catalysts were obtained by linear fitting. The $C_{dl}$ values of the obtained $Ni(OH)_2$, $Ni_5P_4$, $NiSe_2$, and $Ni_5P_4$-Se electrocatalysts are 2.32, 53.69, 28.32, 17.19, and 61.74 mF cm$^{-2}$, respectively (Figure 5). Compared with the single-phase $Ni_5P_4$ and $NiSe_2$, the Se-doped electrocatalyst can expose more active sites to drive the catalysis, which is more conducive to the hydrogen evolution reaction.

## 4. Conclusions

In this work, we successfully prepared a $Ni_5P_4$ catalyst on Ni foam and studied the effect of Se doping on the catalytic performances and structure of single-phase $Ni_5P_4$ catalysts. After characterization, the $Ni_5P_4$-Se displayed a unique porous nanosheet structure, which is beneficial to expose a large quantity of active sites and greatly improve the catalytic performances. The $Ni_5P_4$-Se composite catalyst exhibited amazing catalytic performances, with an overpotential of only 128 mV and a Tafel slope of 163.14 mV dec$^{-1}$ at 10 mA cm$^{-2}$. The overpotential was stabilized at 128 mV for 30 h, demonstrating its superior cycling stability. The direct synthesis of the $Ni_5P_4$-Se on the Ni foam ensured good electrical contact between the catalyst and the conductive support, fast electron transfer, mass transfer and bubble release, thereby endowing more active sites and stability. The strategy of directly

constructing composite catalysts on porous Ni foam provides a new idea for designing efficient water electrolysis catalysts.

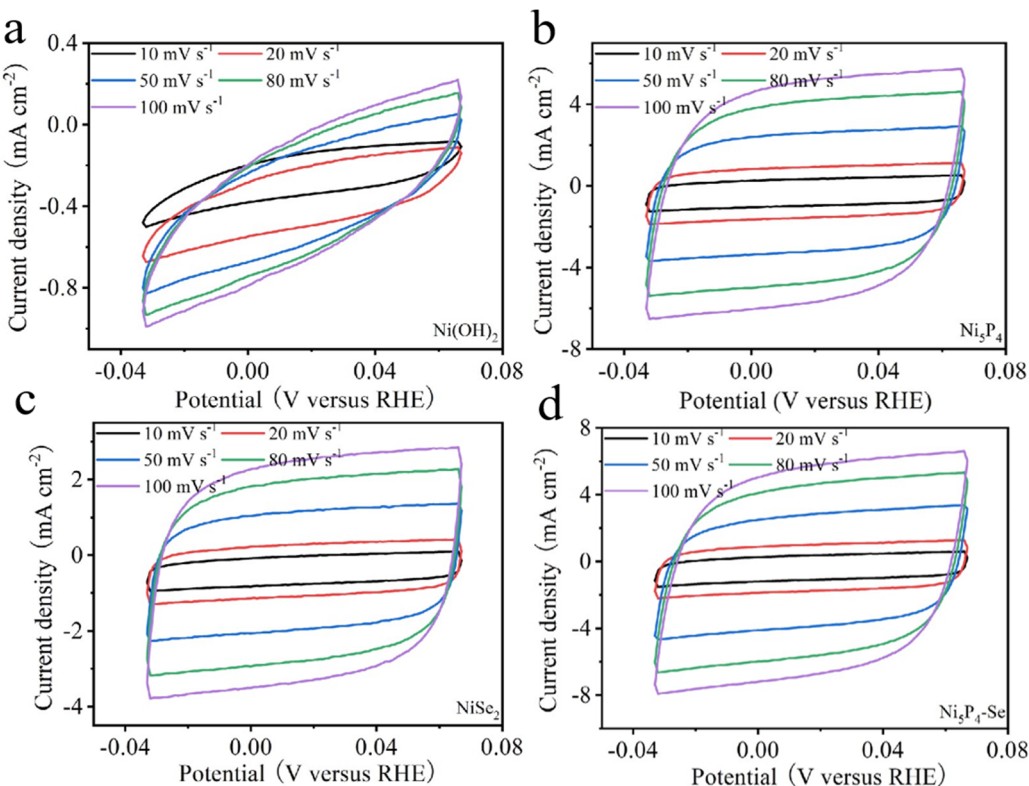

**Figure 4.** CV curves of $Ni(OH)_2$ (**a**), $Ni_5P_4$ (**b**), $NiSe_2$ (**c**), $Ni_5P_4-Se$ (**d**).

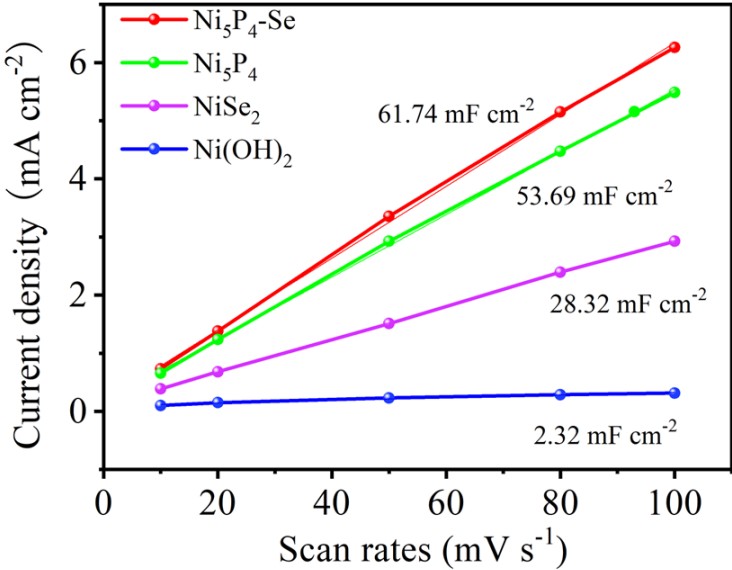

**Figure 5.** The $C_{dl}$ values of $Ni(OH)_2$, $Ni_5P_4$, $NiSe_2$, $Ni_5P_4-Se$.

**Supplementary Materials:** The following supporting information can be downloaded at: https://www.mdpi.com/article/10.3390/catal12091055/s1, Figure S1: XRD patterns of $Ni(OH)_2$ precursors; Figure S2: (a–b) SEM images of $Ni(OH)_2$ precursor; Figure S3: (a–c) SEM images of $Ni_5P_4$; Figure S4: (a–c) SEM images of $NiSe_2$; Figure S5: High-resolution XPS of $Ni_5P_4$ and $NiSe_2$, (a) XPS of Ni 2p in pure $Ni_5P_4$, (b) XPS of P 2p in pure $Ni_5P_4$, (c) XPS of Ni 2p in $NiSe_2$, (d) XPS of Se 3d in $NiSe_2$; Figure S6: TEM images (a-b) of $Ni_5P_4$-Se composites; Figure S7: Energy spectrum for $Ni_5P_4$-Se

composites; Figure S8: SEM image of Ni$_5$P$_4$-Se catalyst after long cycles; Table S1: Comparison of the catalytic activity of catalysts [34–43].

**Author Contributions:** Conceptualization, C.A. and J.L.; methodology, Y.W.; validation, C.A., Y.W. and S.W.; formal analysis, P.J.; investigation, C.A. and P.J.; resources, S.W. and J.L.; data curation, C.A., L.G. and C.Z.; writing—original draft preparation, C.A.; writing—review and editing, S.W. and C.Z.; supervision, N.H.; project administration, S.W. and J.L.; funding acquisition, S.W. All authors have read and agreed to the published version of the manuscript.

**Funding:** This work was supported by the National Natural Science Fund of China (Grant No.: 11632004, 52005151, U1864208), the Research Program of Local Science and Technology Development under the Guidance of Central (216Z4402G), the National Science and Technology Major Project (2017-VII-0011-0106), the Science and Technology Planning Project of Tianjin (20ZYJDJC00030), the Key Program of Research and Development of Hebei Province (202030507040009), the Natural Science Foundation of Hebei Province (E2021202008), the Opening Foundation of State Key Laboratory of Tribology in Tsinghua University (SKLTKF20B03), the Fund for Innovative Research Groups of Natural Science Foundation of Hebei Province (A2020202002) and the Key Project of Natural Science Foundation of Tianjin (S20ZDF077). We also acknowledge support from the "Yuanguang" Scholar Program of Hebei University of Technology.

**Data Availability Statement:** Not applicable.

**Conflicts of Interest:** The authors declare no conflict of interest.

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
