# Peer review of "Se-Doped Ni5P4 Nanocatalysts for High-Efficiency Hydrogen Evolution Reaction"

_catalysts, doi:10.3390/catal12091055_

Round 1
Reviewer 1 Report
The manuscript titled " Se-doped Ni5P4 Nanocatalysts for High-Efficiency Hydrogen Evolution Reaction"
In this manuscript, the authors successfully prepared composites consisted of Se-doped Ni5P4 catalyst anchored on Ni foam and investigated as high-performance photocatalysts for hydrogen evolution reaction. The accomplished samples performance is impressive. Therefore, I would like to recommend published this work after addressing the following points:
1. The Abstract does not contain the main three section, problem, characterization and findings, please rephrase it.
2. The conclusion is also not targeted to the important aspects described in the manuscript; please rephrase it.
3. Introduction is well-organized and well-written, but the importance and novelty of the research should be highlighted and more clearly stated.
4. In experimental section, please provide the purity of your chosen precursors.
5. In characterization section, the conditions used for all characterization techniques should be added.
6. The authors are responsible for the English, which should be polished throughout the manuscript to clear some minor typo/grammar errors.
7. In the introduction part, Some publications are suggested to refer to improve the quality of the manuscript, such as: https://doi.org/10.1016/j.jmrt.2022.03.067, https://doi.org/10.1021/acsomega.1c03735, https://doi.org/10.1007/s10904-022-02389-8.
8. The authors claimed that in cyclic voltammetry curves the potential window were nearly between -0.38 to 0.7 V, Is the negative value of potential window wright, due to potential window should be positive?
9. The author should better improve the beauty and quality of the figures in the manuscript.
11. what about cycle stability of the prepared catalysts?
12. The comparison on catalytic activity of as-prepared samples with some typical catalysts ever reported should be added.
Author Response
- The Abstract does not contain the main three section, problem, characterization and findings, please rephrase it.
Answer: Thank you for your comment! According to your suggestion, the Abstract has been revised, which contain the main three section, problem, characterization and findings. The revised Abstract is as followed.
Abstract: The increasing energy consumption and environmental pollution problems force people to turn their attention to the development and utilization of hydrogen energy, which requires that hydrogen energy can be efficiently prepared. However, the sluggish kinetics of hydrogen evolution reaction (HER) requires higher overpotential. It is urgent to design and fabricate catalysts to drive the procedure and decrease the overpotential of HER. It is well-known that platinum catalysts are the best for HER, but its high cost limits its wide application. Transition metals such as Fe, Co, Mo, and Ni are abundant, and transition metal phosphides are considered as promising HER catalysts. Nevertheless, the catalyst in the powder form is very easily soluble in the electrolyte which leads to the inferior cycling stability. In this work, Ni5P4 anchored on Ni foam is doped with Se powder. After SEM characterization, the Ni5P4-Se are anchored on Ni foam which avoid the using of the conductive additives and binder. The Ni5P4-Se forms a porous nanosheet structure with enhanced electron transfer capability. The prepared Ni5P4-Se exhibits high electrochemical performances. At 10 mA cm-2, the overpotential is only 128 mV and the Tafel slope is 163.14 mV dec-1. And the overpotential is stablized at 128 mV for 30 h, suggesting its excellent cycling stability. The results show that Se doping can make the two phases play a good synergistic effect, which makes the Ni5P4-Se catalyst diaplay excellent HER catalytic activity and stability.
- The conclusion is also not targeted to the important aspects described in the manuscript; please rephrase it.
Answer: Thank you for your suggestion! According to your suggestion, the Conclusion has been revised, which contain the main three section, problem, characterization and findings. The revised Conclusion is as followed.
Conclusion: In this work, we successfully prepared Ni5P4 catalyst on Ni foam, and studied the effect of Se doping on the catalytic performances and structure of single phase Ni5P4 catalysts. After characterization, Ni5P4-Se dispalys a unique porous nanosheet structure, which is beneficial to expose a large quantity of active sites and greatly improve the catalytic performances. The Ni5P4-Se composite catalyst exhibits amazing catalytic performances, with an overpotential of only 128 mV and a Tafel slope of 163.14 mV dec-1 at 10 mA cm-2. The overpotential is stabized at 128 mV for 30 h, demonstrating its superior cycling stability. The direct synthesis of Ni5P4-Se on Ni foam ensures good electrical contact between the catalyst and the conductive support, fast electron transfer, mass transfer, and bubble release, thereby endowing more active sites and stability. The strategy of directly constructing composite catalysts on porous Ni foam provides a new idea for designing efficient water electrolysis catalysts.
- Introduction is well-organized and well-written, but the importance and novelty of the research should be highlighted and more clearly stated.
Answer: Thank you for your comment! According to your suggestion, the Introduction part has been revised in our revised manuscript.
- In experimental section, please provide the purity of your chosen precursors.
Answer: Thanks for your comment! According to the XRD result of the Ni(OH)2 anchored on Ni foam, the content of Ni(OH)2 is about 24 wt%, which have been added in our revised manuscript.
- In characterization section, the conditions used for all characterization techniques should be added.
Answer: Thank you for your suggestion! The conditions used for all characterization techniques have been added in our revised characterization section.
- The authors are responsible for the English, which should be polished throughout the manuscript to clear some minor typo/grammar errors.
Answer: Thanks for your question! According to your suggestion, some grammer and typo mistakes have been revised in our revised manuscript. And the revised manuscript has been checked by our foreign partner, whose native language is English.
- In the introduction part, Some publications are suggested to refer to improve the quality of the manuscript, such as: https://doi.org/10.1016/j.jmrt.2022.03.067, https://doi.org/10.1021/acsomega.1c03735, https://doi.org/10.1007/s10904-022-02389-8.
Answer: Thank you for your suggestion! The above three references have been cited in our revised introduction part (See references 3-5 in our revised manuscript).
- The authors claimed that in cyclic voltammetry curves the potential window were nearly between -0.38 to 0.7 V, Is the negative value of potential window wright, due to potential window should be positive?
Answer: Thank you for your question! We are so sorry that we make a mistake. We have revised them in our revised manuscript.
- The author should better improve the beauty and quality of the figures in the manuscript.
Answer: Thank you for your comment! According to your suggestion, the quality of the figures in our revised manuscript have been improved.
- what about cycle stability of the prepared catalysts?
Answer: Thank you for your question! The long-cycle stability of the Ni5P4-Se electrode was measured at 10 mA cm-2 for 30 h (Figure R1) (See Figure 3d in our revised manuscript). The overpotential was stabilized at 128 mV, demonstrating its excellent cycling durability.
Figure R1 The stability of Ni5P4-Se electrode for 30 h.
- The comparison on catalytic activity of as-prepared samples with some typical catalysts ever reported should be added.
Answer: Thank you for your comment! According to your suggestion, the comparison on catalytic activity of Ni5P4-Se with some reported catalysts has been added in our revised manuscript (See Table R1). We hope this treatment would meet your requirement. Thank you!
Table R1 Comparison of the catalytic activity of catalysts
Catalyst |
Electrolyte |
Overpotential(mV vs.RHE) |
Tafel (mV dec-1) |
Reference |
PNi3S2/NF |
1 M KOH |
137 |
147 |
[1] |
NiCo/Sm2O3 |
1 M KOH |
276 |
162 |
[2] |
Ni2P/LSCN |
0.1 M KOH |
339 |
105 |
[3] |
NiFe LDH@DG |
1 M KOH |
270 |
110 |
[4] |
Ni2P nanofilms |
0.1 M KOH |
315 |
120 |
[5] |
np‐NiFeMoP |
1 M KOH |
223 |
180.3 |
[6] |
FeP |
1 M KOH |
181 |
134 |
[7] |
Cu0.3Co1.7P/NC |
1 M KOH |
220 |
122 |
[8] |
Ni2P |
1 M KOH |
183 |
- |
[9] |
CoNiP‐1:1 NWs |
1 M KOH |
252 |
128 |
[10] |
Ni5P4-Se |
1 M KOH |
128 |
163.14 |
This work |
References
- Ding, Y. H.; Li, H. Y.; Hou, Y., Phosphorus-doped nickel sulfides/nickel foam as electrode materials for electrocatalytic water splitting. Int. J. Hydrog. Energy 2018, 43, 19002-19009.
- Liu, H. H.; Zeng, S.; He, P.; Dong, F. Q.; He, M. Q.; Zhang, Y.; Wang, S.; Li, C. X.; Liu, M. Z.; Jia, L. P., Samarium oxide modified Ni-Co nanosheets based three-dimensional honeycomb film on nickel foam: A highly efficient electrocatalyst for hydrogen evolution reaction. Electrochim. Acta 2019, 299, 405-414.
- Wang, Y. R.; Wang, Z. J.; Jin, C.;Li, C.;Li, X. W.; Li, Y. F.; Yang, R. Z.; Liu, M. L., Enhanced overall water electrolysis on a bifunctional perovskite oxide through interfacial engineering. Electrochim. Acta 2019, 318, 120-129.
- Jia, Y.; Zhang, L. Z.; Gao, G. P.; Chen, H.; Wang, B.; Zhou, J. Z.; Soo, M. T.; Hong, M.; Yan, X. C.; Qian, G. R.; Zou, J.; Du, A. J.; Yao, X. D., A Heterostructure Coupling of Exfoliated Ni-Fe Hydroxide Nanosheet and Defective Graphene as a Bifunctional Electrocatalyst for Overall Water Splitting. Adv. Mater. 2017, 29, 8.
- Li, Y.; Cai, P. W.; Ci, S. Q.; Wen, Z. H., Strongly Coupled 3D Nanohybrids with Ni2P/Carbon Nanosheets as pH-Universal Hydrogen Evolution Reaction Electrocatalysts. ChemElectroChem 2017, 4, 340-344.
- Qian, H. X.; Li, K. Y.; Mu, X. B.; Zou, J. Z.; Xie, S. H.; Xiong, X. B.; Zeng, X. R., Nanoporous NiFeMoP alloy as a bifunctional catalyst for overall water splitting. Int. J. Hydrog. Energy 2020, 45, 16447-16457.
- Lv, C. C.; Peng, Z.; Zhao, Y. X.; Huang, Z. P.; Zhang, C., The hierarchical nanowires array of iron phosphide integrated on a carbon fiber paper as an effective electrocatalyst for hydrogen generation. J. Mater. Chem. A 2016, 4, 1454-1460.
- Song, J. H.; Zhu, C. Z.; Xu, B. Z.;Fu, S. F.; Engelhard, M. H.; Ye, R. F.; Du, D.; Beckman, S. P.; Lin, Y. H., Bimetallic Cobalt-Based Phosphide Zeolitic Imidazolate Framework: CoPx Phase-Dependent Electrical Conductivity and Hydrogen Atom Adsorption Energy for Efficient Overall Water Splitting. Adv. Energy Mater. 2017, 7, 9.
- Read, C. G.; Callejas, J. F.; Holder, C. F.; Schaak, R. E., General Strategy for the Synthesis of Transition Metal Phosphide Films for Electrocatalytic Hydrogen and Oxygen Evolution. ACS Appl. Mater. Interfaces 2016, 8, 12798-12803.
- Amorim, I.; Xu, J. Y.; Zhang, N.;Xiong, D. H.; Thalluri, S. M.; Thomas, R.; Sousa, J. P. S.; Araujo, A.; Li, H.; Liu, L. F., Bi-metallic cobalt-nickel phosphide nanowires for electrocatalysis of the oxygen and hydrogen evolution reactions. Catal. Today 2020, 358, 196-202.

Reviewer 2 Report
In this study, the authors have prepared an electrocatalyst for hydrogen evolution reaction based on metallic nickel phosphide (Ni5P4 ). The present catalyst, Se-doped Ni5P4 nanocatalyst on nickel foam (Ni5P4-Se), demonstrates high electrochemical performance for the hydrogen evolution reaction (HER). It requires an overpotential of 128 mV for HER at a current density of 10 mA cm-2 . However, there are some key issues that are not explained clearly, and the data are not sufficient to support the author’s viewpoint. Therefore, the reviewer considers that the current manuscript is not ready for publication, but if the author could improve the science and novelty, it might be suitable for this journal. The following comments might need to be considered before making a final decision.
1. The novelty of the present work should be clearly specified in the introduction section. The novelty of the research work is very important. The abstract needs to be modified.
2. The authors synthesized the materials by combining hydrothermal and post-phosphorization. Then the electrochemical property of the samples has been estimated considering its polarization curve, EIS, and long stability estimation. The comments are that what is the science, mechanism? It might be better than the authors providing an insightful discussion.
3. There is much literature reporting the development of electrocatalysis, how does the performance level compare with previous literature?
4. In terms of performance, despite having better HER performance, the overpotential for 10 mA cm -2 of HER is 128 mV vs RHE, which is not comparable to that of the reported collector self-supported electrocatalysts. In my opinion, this is not enough to show that it is a good catalyst for HER.
5. The resolution ratio of the Figure is very low. For example, in addition, please refit the XPS spectra P and Se.
6. The scale bars in Figure 2d-f should be included.
7. To confirm the micromorphology and the respective lattice fringes, the authors should provide the TEM analysis of Ni 5P 4-Se.
8. For the EIS spectra, how about the electrode areas for the different electrodes? Different electrode areas would result in resistance difference.
9. Similar research on the electrocatalyst can be cited in the appropriate positions for the reference of data presentation and explanation. Materials Today Nano, 17, March 2022, 100146, Composites Part B: Engineering, Volume 239, 15 June 2022, 109992.
10. To show the better stability of the as-synthesized catalyst, provide the more than 24 h chronopotentiometry test.
11. The surface reconstruction in HER after stability was not discussed.
12. The controlled selenium doping in the transitional metal phosphide and its characterization can be referenced from the ACS Appl. Energy Mater. 2021, 4, 1, 404–415 with proper citation.
Author Response
- The novelty of the present work should be clearly specified in the introduction section. The novelty of the research work is very important. The abstract needs to be modified.
Answer: Thank you for your suggestion! According to your comments, the abstract and the introduction parts have been revised in our revised manuscript.
- The authors synthesized the materials by combining hydrothermal and post-phosphorization. Then the electrochemical property of the samples has been estimated considering its polarization curve, EIS, and long stability estimation. The comments are that what is the science, mechanism? It might be better than the authors providing an insightful discussion.
Answer: Thank you very much for your question! The abstract, Introduction and conclusion parts have been revised in our revised manuscript.
- There is much literature reporting the development of electrocatalysis, how does the performance level compare with previous literature?
Answer: Thanks for your question! According to your suggestion, the comparison on catalytic activity of Ni5P4-Se with some reported catalysts has been added in our revised manuscript (See Table R1).
Table R1 Comparison of the catalytic activity of catalysts
Catalyst |
Electrolyte |
Overpotential(mV vs.RHE) |
Tafel (mV dec-1) |
Reference |
PNi3S2/NF |
1 M KOH |
137 |
147 |
[1] |
NiCo/Sm2O3 |
1 M KOH |
276 |
162 |
[2] |
Ni2P/LSCN |
0.1 M KOH |
339 |
105 |
[3] |
NiFe LDH@DG |
1 M KOH |
270 |
110 |
[4] |
Ni2P nanofilms |
0.1 M KOH |
315 |
120 |
[5] |
np‐NiFeMoP |
1 M KOH |
223 |
180.3 |
[6] |
FeP |
1 M KOH |
181 |
134 |
[7] |
Cu0.3Co1.7P/NC |
1 M KOH |
220 |
122 |
[8] |
Ni2P |
1 M KOH |
183 |
- |
[9] |
CoNiP‐1:1 NWs |
1 M KOH |
252 |
128 |
[10] |
Ni5P4-Se |
1 M KOH |
128 |
163.14 |
This work |
References
- Ding, Y. H.; Li, H. Y.; Hou, Y., Phosphorus-doped nickel sulfides/nickel foam as electrode materials for electrocatalytic water splitting. Int. J. Hydrog. Energy 2018, 43, 19002-19009.
- Liu, H. H.; Zeng, S.; He, P.; Dong, F. Q.; He, M. Q.; Zhang, Y.; Wang, S.; Li, C. X.; Liu, M. Z.; Jia, L. P., Samarium oxide modified Ni-Co nanosheets based three-dimensional honeycomb film on nickel foam: A highly efficient electrocatalyst for hydrogen evolution reaction. Electrochim. Acta 2019, 299, 405-414.
- Wang, Y. R.; Wang, Z. J.; Jin, C.;Li, C.;Li, X. W.; Li, Y. F.; Yang, R. Z.; Liu, M. L., Enhanced overall water electrolysis on a bifunctional perovskite oxide through interfacial engineering. Electrochim. Acta 2019, 318, 120-129.
- Jia, Y.; Zhang, L. Z.; Gao, G. P.; Chen, H.; Wang, B.; Zhou, J. Z.; Soo, M. T.; Hong, M.; Yan, X. C.; Qian, G. R.; Zou, J.; Du, A. J.; Yao, X. D., A Heterostructure Coupling of Exfoliated Ni-Fe Hydroxide Nanosheet and Defective Graphene as a Bifunctional Electrocatalyst for Overall Water Splitting. Adv. Mater. 2017, 29, 8.
- Li, Y.; Cai, P. W.; Ci, S. Q.; Wen, Z. H., Strongly Coupled 3D Nanohybrids with Ni2P/Carbon Nanosheets as pH-Universal Hydrogen Evolution Reaction Electrocatalysts. ChemElectroChem 2017, 4, 340-344.
- Qian, H. X.; Li, K. Y.; Mu, X. B.; Zou, J. Z.; Xie, S. H.; Xiong, X. B.; Zeng, X. R., Nanoporous NiFeMoP alloy as a bifunctional catalyst for overall water splitting. Int. J. Hydrog. Energy 2020, 45, 16447-16457.
- Lv, C. C.; Peng, Z.; Zhao, Y. X.; Huang, Z. P.; Zhang, C., The hierarchical nanowires array of iron phosphide integrated on a carbon fiber paper as an effective electrocatalyst for hydrogen generation. J. Mater. Chem. A 2016, 4, 1454-1460.
- Song, J. H.; Zhu, C. Z.; Xu, B. Z.;Fu, S. F.; Engelhard, M. H.; Ye, R. F.; Du, D.; Beckman, S. P.; Lin, Y. H., Bimetallic Cobalt-Based Phosphide Zeolitic Imidazolate Framework: CoPx Phase-Dependent Electrical Conductivity and Hydrogen Atom Adsorption Energy for Efficient Overall Water Splitting. Adv. Energy Mater. 2017, 7, 9.
- Read, C. G.; Callejas, J. F.; Holder, C. F.; Schaak, R. E., General Strategy for the Synthesis of Transition Metal Phosphide Films for Electrocatalytic Hydrogen and Oxygen Evolution. ACS Appl. Mater. Interfaces 2016, 8, 12798-12803.
- Amorim, I.; Xu, J. Y.; Zhang, N.;Xiong, D. H.; Thalluri, S. M.; Thomas, R.; Sousa, J. P. S.; Araujo, A.; Li, H.; Liu, L. F., Bi-metallic cobalt-nickel phosphide nanowires for electrocatalysis of the oxygen and hydrogen evolution reactions. Catal. Today 2020, 358, 196-202.
- In terms of performance, despite having better HER performance, the overpotential for 10 mA cm-2 of HER is 128 mV vs RHE, which is not comparable to that of the reported collector self-supported electrocatalysts. In my opinion, this is not enough to show that it is a good catalyst for HER.
Answer: Thanks for your suggestion! The comparison on catalytic activity of Ni5P4-Se with some reported catalysts has been added in our revised manuscript (See Table 1). The overpotential of the Ni5P4-Se catalyst is not very low, which can be stable for 30 h, demonstrating its excellent cycling stability.
- The resolution ratio of the Figure is very low. For example, in addition, please refit the XPS spectra P and Se.
Answer: Thanks for your suggestion! According to your suggestion, the quality of the figures in our revised manuscript have been improved. And the XPS spectra P and Se have been refitted (See Figure 1c and d).
- The scale bars in Figure 2d-f should be included.
Answer: Thanks for your suggestion! The scale bars in Figure 2d-f have been added in our revised manuscript.
- To confirm the micromorphology and the respective lattice fringes, the authors should provide the TEM analysis
Answer: Thanks for your suggestion! The TEM images of the Ni5P4-Se catalysts (See Figure S6) have been added in our revised manuscript.
Figure R2 TEM images (a-b) of Ni5P4-Se composites.
- For the EIS spectra, how about the electrode areas for the different electrodes? Different electrode areas would result in resistance difference.
Answer: Thanks for your question! In our electrochemical tests, the electrode areas for different electrodes is 1 cm2, which has been added in our revised manuscript.
- Similar research on the electrocatalyst can be cited in the appropriate positions for the reference of data presentation and explanation. Materials Today Nano, 17, March 2022, 100146, Composites Part B: Engineering, Volume 239, 15 June 2022, 109992.
Answer: Thanks for your suggestion! The reference “Materials Today Nano, 2022, 17, 100146” and “Composites Part B: Engineering, 2022, 239, 109992” have been cited in our revised manuscript (See reference 6 and 8).
- To show the better stability of the as-synthesized catalyst, provide the more than 24 h chronopotentiometry test.
Answer: Thanks for your suggestion! The long-cycle stability of the Ni5P4-Se electrode was measured at 10 mA cm-2 for 30 h (Figure R3) (See Figure 3d in our revised manuscript). The overpotential was stabilized at 128 mV, demonstrating its excellent cycling durability.
Figure R3 The stability of Ni5P4-Se electrode for 30 h.
- The surface reconstruction in HER after stability was not discussed.
Answer: Thanks for your suggestion! The SEM image of Ni5P4-Se catalyst has been added in our revised supporting information (See Figure S8). As we can seen the Ni5P4-Se catalyst exhibits nanosheets structure, which maintains its original morphological structure, further demonstrating its structural stability.
Figure R4 SEM image of Ni5P4-Se catalyst after long cycles.
- The controlled selenium doping in the transitional metal phosphide and its characterization can be referenced from the ACS Appl. Energy Mater. 2021, 4, 1, 404–415 with proper citation.
Answer: Thanks for your comment! The reference “ACS Appl. Energy Mater. 2021, 4(1), 404-415” has been cited in our revised manuscript (See reference 26). We hope this treatment would meet your requirement. Thank you!

Round 2
Reviewer 1 Report
The manuscript is acceptable in the present form since authors have well addressed the questions proposed by referees.
Reviewer 2 Report
All the comments and suggstion are appended carfully.